# Human Complement Inhibits Myophages against *Pseudomonas aeruginosa*

**DOI:** 10.3390/v15112211

**Published:** 2023-11-03

**Authors:** Julia E. Egido, Simon O. Dekker, Catherine Toner-Bartelds, Cédric Lood, Suzan H. M. Rooijakkers, Bart W. Bardoel, Pieter-Jan Haas

**Affiliations:** 1Department of Medical Microbiology, University Medical Center Utrecht, 3584 CX Utrecht, The Netherlands; 2Laboratory of Gene Technology, Department of Biosystems, Katholieke Universiteit Leuven, B-3000 Leuven, Belgium; 3Centre of Microbial and Plants Genetics, Department of Microbial and Molecular Systems, Katholieke Universiteit Leuven, B-3000 Leuven, Belgium

**Keywords:** phage therapy, complement system, *Pseudomonas aeruginosa*

## Abstract

Therapeutic bacteriophages (phages) are primarily chosen based on their in vitro bacteriolytic activity. Although anti-phage antibodies are known to inhibit phage infection, the influence of other immune system components is less well known. An important anti-bacterial and anti-viral innate immune system that may interact with phages is the complement system, a cascade of proteases that recognizes and targets invading microorganisms. In this research, we aimed to study the effects of serum components such as complement on the infectivity of different phages targeting *Pseudomonas aeruginosa*. We used a fluorescence-based assay to monitor the killing of *P. aeruginosa* by phages of different morphotypes in the presence of human serum. Our results reveal that several myophages are inhibited by serum in a concentration-dependent way, while the activity of four podophages and one siphophage tested in this study is not affected by serum. By using specific nanobodies blocking different components of the complement cascade, we showed that activation of the classical complement pathway is a driver of phage inhibition. To determine the mechanism of inhibition, we produced bioorthogonally labeled fluorescent phages to study their binding by means of microscopy and flow cytometry. We show that phage adsorption is hampered in the presence of active complement. Our results indicate that interactions with complement may affect the in vivo activity of therapeutically administered phages. A better understanding of this phenomenon is essential to optimize the design and application of therapeutic phage cocktails.

## 1. Introduction

Over the last decades, the increase in multi-drug resistant bacteria has spurred a renewed interest in bacteriophages (phages) as an alternative to antibiotics [1]. Phages used in therapy predominantly follow a lytic cycle, which means that they can rapidly kill infected bacteria and propagate by spreading to neighboring cells [2]. These phages belong in the majority of cases to the order Caudovirales, and are morphologically characterized by an icosahedral capsid, or head, which contains the genetic material, and a tail. The tail can further determine the morphotype of the phage. Phages with contractile tails are named myophages, phages with short tails are named podophages, and phages with long, non-contractile tails are named siphophages. In contrast to most antibiotics, phages have the advantage of being very specific in the bacteria that they can target. However, this also means that phage therapy often has to be highly tailored to the needs of each patient. Therapeutic success may therefore depend on using the most optimal phages to treat each infection [3].

A pathogen for which phage therapy is particularly relevant is *Pseudomonas aeruginosa*. This species of Gram-negative bacteria is part of the ESKAPE list, a group of pathogens that are the leading cause of nosocomial infections worldwide. It is an opportunistic pathogen that causes infections ranging from burn wound complications to pneumonia and severe sepsis. It is particularly worrisome for immunocompromised patients with underlying conditions, such as cystic fibrosis or hematological malignancies. Antibiotic resistance in *P. aeruginosa* is rising rapidly, making it a prime candidate for treatment with phage therapy [4]. The potential of phage therapy against *P. aeruginosa* infections is showcased in recent case reports [5,6,7]. Several clinical studies have also been carried out to evaluate the safety of phage therapy in humans against this pathogen, with positive outcomes [8,9]. Efficacy is also assessed in some of these studies, although the results can be difficult to interpret because of the challenges in study design [10,11]. Nonetheless, most of the evidence comes from clinical cases and case series, with the number of randomized clinical trials still being limited.

There are several reasons why the efficacy of phage therapy is still disputed. Bacteria can evade phage infection via several defense mechanisms, and they can rapidly mutate to develop resistance to phages [12,13]. To prevent this, mixes of different phages, or cocktails, are often administered. However, finding several phages to target a pathogenic strain can be challenging, and the optimization of phage cocktail compositions is not always straightforward [14]. Another major caveat is that the patient’s immune system may recognize and target phages as exogenous organisms. This has been shown to occur in different ways, such as the induction of neutralizing antibodies [15] or phagocytosis of phage virions by immune cells [16]. Depending on the patient and the route of administration, the immune response could lead to inactivation of the phages before they can reach their goal.

A major part of the immune system that phages encounter when entering the blood and tissues is the complement system. The complement system mediates innate immunity to exogenous entities, including pathogens. It consists of a cascade of proteases that can activate via different pathways: the classical pathway is activated through the action of antibodies, the alternative pathway creates an amplification loop that increases activation, and the lectin pathway is triggered by recognition of molecular patterns [17]. Following activation, the pathways converge at the formation of a C3 convertase, which cleaves C3 causing C3b to deposit on the bacterial surface. C3b can act as an opsonin, triggering and promoting phagocytosis by immune cells. The presence of C3b also causes the C3 convertase to switch specificity to recognize C5. Eventually, this leads to formation of the membrane attack complex (MAC), which can directly lyse Gram-negative bacteria [18]. Although complement could in some instances act synergistically with phage therapy [19], there is also evidence that complement can recognize viral particles [20], and even phages [21].

Our goal was to study the effect of human serum, in particular of the complement system, on lytic phage activity. To address this, we assessed different *P. aeruginosa* phages in the presence of serum. Our results reveal that serum inhibits the lytic activity of certain phages, predominantly myophages. To study the involvement of the complement system in this phenomenon, we used nanobodies inhibiting specific complement proteins. In this way, we have demonstrated that the recognition of phages by early components of the complement cascade can impair their binding to bacteria. The fact that different phages perform differently when challenged by the innate immune system could have important repercussions in the way that we select phages for therapy.

## 2. Materials and Methods

### 2.1. Phages and Strains

Phage PBJ was isolated in our lab after two rounds of amplification from a mixed stock using *P. aeruginosa* strain PAO1 as the host. Original stocks of *Pseudomonas* phages 14-1, LKD16, and LUZ19 were kindly provided by Rob Lavigne (KU Leuven, Belgium). *Pseudomonas* phages (Pa collection) and phage K were obtained from the Fagenbank (Delft, The Netherlands). Phage amplification was carried out by infecting the host strain PAO1 overnight at 37 °C in Lysogeny Broth (LB). Bacterial debris was removed by centrifugation at 11,000× *g* (RCF) for 40 min at 4 °C. Phages were then incubated for 2 h on ice with a solution of 10% PEG-8000 and 0.5 M NaCl and precipitated by centrifugation at 11,000× *g* (RCF) for 40 min at 4 °C. The preparation was mixed with chloroform, after which the aqueous phase was sterilized using a 0.2 μm filter and incubated with DNase and RNase (5 μg/mL each) for 30 min at room temperature. Final purity was achieved by filtration through a Zeba Spin desalting column (40K MWCO, Thermo Fisher Scientific, Waltham, MA, USA). Phages were recovered and stored in SM buffer (100 mM NaCl, 8 mM MgSO_4_·7H_2_O, 50 mM Tris-Cl).

Phages labeled with an azido-handle were produced in a PAO1 metZ knockout strain, obtained from the University of Washington collection [22]. Phage amplification was performed by infecting the bacteria overnight in M9 minimal medium supplemented with methionine (5 mg/L) and the methionine homolog azido-homo-alanine (40 mg/L). After removal of bacterial remains, phages were pelleted using PEG-8000/NaCl as described above, filter-sterilized, and concentrated using an Amicon filter (100 kDa cut-off) (Sigma-Aldrich, St. Louis, MO, USA). To achieve fluorescent labeling, phages were incubated with DBCO-AF488 (Jena Bioscience, Jena, Germany) for 1 h at 37 °C. Excess dye was washed off by washing with SM buffer in an Amicon filter (100 kDa cut-off).

PAO1 expressing sfCherry was obtained by transforming PAO1 with the plasmid pUCP30T modified to express sfCherry [23]. Bacteria were made competent by washing in 300 mM sucrose, after which the plasmid was introduced by electroporation. Selection was performed on plates containing gentamycin (30 μg/mL). *S. aureus* strain ATCC 19685 was kindly provided by Ana Rita Costa (Fagenbank; TU Delft, The Netherlands).

### 2.2. Genome Sequencing and Bioinformatics Analysis

The genomic DNA library for phage PBJ was prepared with the Nextera Flex kit (Illumina, San Diego, CA, USA) and sequenced using the Illumina MiniSeq using a paired-end approach (2 × 150 bp). The reads were then controlled for quality using FastQC v0.12.0 [24] and we used Trimmomatic [25] to remove adapter sequences, filter by length (>50 bp), and trim lower quality regions (Trimmomatic options: ILLUMINACLIP:NexteraPE-PE.fa:2:30:10LEADING:3 TRAILING:3 SLIDINGWINDOW:4:15 MINLEN:50). The genome was assembled de novo using the SPAdes assembler with default options, including the pipeline option “--careful” [26]. The quality of the assembly was inspected with Bandage [27] to allow us to extract the relevant circular phage contigs. The genome of phage PBJ was annotated using Pharokka [28] and compared with the closely related phage 14-1 genome (NC_011703.1).

### 2.3. Bacterial Growth

For all experiments, bacteria were cultured overnight in LB from a single colony at 37 °C with shaking. On the day of the experiment, bacteria were diluted 1:31 in LB and sub-cultured to mid-log phase (OD_600nm_~0.5). Then, bacteria were washed in RPMI 1640 buffer (Thermo Fisher, Waltham, MA, USA), supplemented with 0.05% human serum albumin, Sanquin), pelleted, and resuspended to an OD_600nm_ of 1.0 (~8 × 10^8^ bacteria/mL) in RPMI buffer. Bacteria were further diluted as specified for the different assays.

### 2.4. Serum Preparation and Reagents

Human pooled serum (HPS) was isolated from healthy volunteers at the UMC Utrecht (The Netherlands). Briefly, blood was drawn, allowed to clot, and centrifuged to isolate the serum. Serum from 15–20 donors was pooled, aliquoted, and stored at −80 °C. Heat-inactivated HPS (HI HPS) was prepared by treating HPS at 56 °C for 30 min to selectively inactivate complement proteins. Pooled IgG/IgM was isolated from HPS as previously described [29]. Compstatin was kindly provided by John Lambris (University of Pennsylvania, Philadelphia, PA, USA). OMCI was produced and purified as previously described [30]. Complement-targeting nanobodies were produced as described by their developers [31,32,33,34]. Monoclonal mouse antibody bH6 recognizing human activated C3 (Hycult Biotech, Uden, The Netherlands) was produced as previously described [18].

### 2.5. Microplate Reader Assays

Bacteria (PAO1), phages, and SYTOX Green nucleic acid stain (Thermo Fisher, Waltham, MA, USA) were mixed together to a final concentration of ~2 × 10^7^ bacteria/mL and 5 μM SYTOX Green. Concentration of phages is dependent on the multiplicity of infection (MOI), which is indicated for each experiment. Where indicated, human pooled serum, heat-inactivated human pooled serum, purified IgG/IgM from pooled serum, or serum with complement inhibitors was added in the concentrations specified in each of the figures. For experiments with pre-opsonized bacteria, prior to the addition of phages, bacteria were incubated with serum at 37 °C for 30 min, centrifuged at 2000–3000× *g* (RCF) for 8 min, and resuspended in fresh medium. Fluorescence measurements were performed in a microplate reader (FLUOstar Omega, BMG Labtech, Ortenberg, Germany) at 37 °C, in a clear, flat-bottomed 96-well plate (Corning, Corning, NY, USA), with the following settings: λ_excitation_ = 490 nm, bandwidth = 14 nm; λ_emission_ = 537 nm, bandwidth 30 nm; gain = 1300.

### 2.6. Determination of Bacterial Viability

Bacteria (*S. aureus* strain ATCC 19685) were mixed with phage K to a concentration of 2 × 10^7^ bacteria/mL and a phage concentration of 2 × 10^8^ phages/mL (MOI of 10). The mixture was incubated at 37 °C with shaking for 120 min. Samples were taken and serial dilutions were performed in RPMI buffer. A total of 5 µL of each sample was plated on LB agar and incubated overnight at 37 °C. Colonies were counted and the CFUs/mL was calculated.

### 2.7. Flow Cytometry

Bacteria (PAO1-sfCherry) at a final concentration of ~2 × 10^7^ bacteria/mL were mixed with PBJ-AF488 in various MOIs and, in some cases, 10% HPS. Nanobodies (anti-C1q and anti-C3) were added when indicated at 1 μM. Heat-inactivated PBJ-AF488 (heat treatment: 85 °C for 50 min) was used as a control at an MOI of 200. Samples were incubated for 5 min at 37 °C with shaking, after which bacteria were washed in cold buffer and pelleted at 4000 rpm at 4 °C for 8 min. The supernatant was removed, and bacteria were resuspended in a solution of bH6-AF405 antibody against C3b at 3 μg/mL in RPMI 1640 with 0.05% HAS. After incubation on ice for 10 min, samples were fixed using cold PFA to a final concentration of 1%. Samples were finally pelleted at 3500 rpm for 8 min, resuspended in fresh buffer to a concentration between 10^6^ and 10^7^ bacteria/mL, and analyzed via flow cytometry (MACS Quant, Miltenyi Biotech, Bergisch Gladbach, Germany). Per condition, 10 μL was measured. Bacteria were gated using sfCherry signal height.

### 2.8. Widefield Fluorescence Microscopy

Bacteria (PAO1-sfCherry) were grown to mid-log phase (OD_600nm_~0.5) in LB, pelleted, and resuspended to an OD_600nm_ of 1.0 (~8 × 10^8^ bacteria/mL) in RPMI 1640 with 0.05% HSA. Bacteria at a final concentration of 10^8^ bacteria/mL were mixed with phages at an MOI of 50. HPS was added to some conditions at a concentration of 10%. The mixture was incubated for 5 min at 37 °C with shaking. Then, samples were fixed using cold PFA to a final concentration of 1%. After 15 min incubation on ice, the samples were centrifuged at 3500 rpm for 8 min. The supernatant was removed, and the samples were resuspended in fresh RPMI to an estimated concentration of 4 × 10^8^ bacteria/mL. Then, the samples were immobilized on microscope slides prepared with agarose pads. To make the agarose pads, 50 μL 1% agarose gel in PBS was added to an objective slide, after which a siliconized cover slip was applied on top of the agarose gel to flatten it. Once the agarose was solid, the cover slips were removed and 10 μL of the sample was added to the pad. After the samples were dry, a cover slip was placed on the pad. The samples were imaged using a Leica TCS SP5 II microscope with an HC PL APO CS 100×/1.4 OIL objective (Leica Microsystems, Amsterdam, The Netherlands). The images were obtained by overlaying the modes phase contrast and widefield fluorescence with cube filters for RFP (N21, dichroic mirror: 580 nm) and GFP (dichroic mirror: 500 nm), respectively.

### 2.9. C1q ELISA

A Nunc MaxiSorp ELISA plate (Thermo Fisher Scientific, Waltham, MA, USA) was coated overnight at 4 °C with the following: 5 × 10^8^ phage PBJ per well in PBS, 50 ng per well of an anti-*S. aureus* WTA human IgM antibody in PBS as a positive control, or PBS as negative control. To prepare the phages for this purpose, a sterile lysate of phage PBJ was concentrated by ultracentrifugation and purified of endotoxins using EndoTrap HD (Lionex GmbH, Braunschweig, Germany) according to the manufacturer’s instructions. After coating, the plate was blocked with 4% skimmed milk powder (Campina, Wolvega, The Netherlands). Next, C1q (Complement Technology, Tyler, TX, USA) was added in a range of concentrations. To detect C1q binding, polyclonal rabbit anti-human C1q antibody (Agilent Technologies, Santa Clara, CA, USA) was used at 1 µg/mL. This was in turn detected using goat anti-rabbit IgG antibody conjugated with horseradish peroxidase (Southern Biotech, Birmingham, AL, USA) diluted to 1:5000. Before each of the steps, the plate was washed 3 times with PBS supplemented with 0.05% Tween-20 (PBS-T). Finally, the substrate tetramethylbenzidine (TMB) was added and incubated for 3 min, after which the reaction was stopped with H_2_SO_4_ (0.5 M). The detection antibodies as well as C1q were diluted in PBS-T supplemented with 1% skimmed milk powder. All of the incubation steps were performed at room temperature for 1 h unless otherwise stated. Absorbance was measured at 450 nm using an iMark Microplate Reader (Bio-Rad, Hercules, CA, USA).

### 2.10. Data Analysis and Statistical Testing

Data visualization and statistical analyses were performed in GraphPad Prism 9 and are further specified in the figure legends. Flow cytometry data were analyzed using FlowJo™ v10.8.1 software. Widefield fluorescence microscopy images were processed using Fiji. Figures were produced using Adobe Illustrator.

## 3. Results

### 3.1. Human Serum Impairs Activity of Myophages PBJ and 14-1

We studied the effect of human serum on the activity of phages against the serum-resistant *P. aeruginosa* strain PAO1. We evaluated four different virulent phages: myophages PBJ and 14-1 and podophages LKD16 and LUZ19. PBJ is a Pbunavirus that belongs to the Pbunavirus pv141 species with 96% sequence identity over 96% of the genome of phage 14-1. Phages 14-1 (species: Pbunavirus pv141), LKD16 (species: Phikmvvirus LKD16), and LUZ19 (species: Phikmvvirus LUZ19) have been extensively characterized in the literature [35,36,37,38]. The phages were combined in different multiplicities of infection (MOIs) with PAO1 in the presence of different concentrations of human pooled serum (HPS). Phage-mediated killing was monitored using the fluorescent DNA dye assay as previously described [39]. Briefly, we used the membrane-impermeant DNA dye SYTOX Green, which can only stain bacterial DNA once the cells are damaged or lysed. This assay allows us to monitor phage-mediated damage in real time, where an increase in fluorescence correlates with bacterial killing. When comparing all four phages, we found that the addition of HPS inhibited damage induction for the myophages, resulting in a much slower increase of the fluorescence signal (Figure 1a,b). In contrast, no differences in activity were observed for the podophages in the presence or absence of HPS. Based on these results, we used the time of phage-mediated damage induction as a readout for phage activity to compare all conditions more easily. In this way, a shorter time indicates a more rapid or efficient phage infection. We confirmed that damage induction by phage PBJ was delayed with the addition of HPS. The effect was concentration-dependent, where a higher concentration of HPS was needed to inhibit a higher phage MOI (Figure 1c). Phage 14-1 presented a similar pattern, where the lower MOIs (0.1 and 1) showed clear inhibition by HPS (Figure 1d). However, this phage seemed slightly less sensitive to HPS than PBJ, as its activity was largely unaffected at the highest MOI. Strikingly, the activity of podophages LKD16 and LUZ19 did not change with the addition of HPS (Figure 1e,f). These observations indicate that serum can selectively inhibit certain phages in a concentration-dependent manner, while other phages are not sensitive to inhibition by serum.

### 3.2. Serum-Mediated Inhibition Is Found for a Broader Set of Myophages

To further determine whether the effect is specific to the phage morphotype, we evaluated the activity of 9 more myophages, 2 podophages, and 1 siphophage in the presence and absence of human serum. These were uncharacterized phages initially isolated from sewage water and classified according to tail morphology [40]. Phage activity was assessed using the fluorescent DNA dye assay. In the absence of serum, we observed differences in the killing kinetics of the different phages, with latent periods ranging from around 30 to over 100 min (Appendix A). When serum was added, time of damage induction was greatly increased for 7 out of the 9 myophages (Figure 2a and Appendix A). In contrast, the activity of myophages Pa33 and Pa36, podophages Pa18 and Pa29, and siphophage Pa28 was not affected by the addition of serum. Notably, siphophage Pa28 and podophage Pa29 were not very active in the absence of serum, which could be due to PAO1 not being the optimal host for this phage. These observations suggest that some myophages, if not all, could be more sensitive to serum-mediated inhibition. However, given that these phages and their receptors have not been characterized, it could be that the effect is not dependent on the morphotype, but rather due to the phages being closely related or targeting the same receptor. To rule this out, we tested whether serum can also inhibit phage K, a myophage targeting *Staphylococcus aureus* that uses wall teichoic acid (WTA) as a receptor. To do so, we incubated bacteria with the phage in the presence or absence of serum and examined bacterial viability through colony counting. Indeed, no reduction in colony-forming units (CFUs) was detected when phages were added in the presence of serum (Figure 2b). As phage K targets a receptor absent on *P. aeruginosa*, it seems that inhibition by serum is not dependent on the phage receptor. These results further suggest that myophages are more sensitive to inhibition by human serum.

### 3.3. Phage Inhibition in Serum Is Mediated by the Early Stages of the Complement Cascade

Next, we investigated the components in serum responsible for the inhibitory effect. We first aimed to rule out whether this effect was due to the potential presence of neutralizing anti-phage antibodies in our serum pool. For this, we combined phages with HPS, HPS with heat-inactivated complement (HI HPS), or IgG and IgM purified from HPS, and studied killing kinetics with the fluorescent DNA dye assay (Figure 3a). Phages performed equally well in the presence of antibodies and heat-inactivated serum as they did in the control condition. As purified antibodies and serum with inactive complement did not have an effect on phage activity, we inferred that the complement system could be driving phage inhibition. In all complement activation pathways, cleavage of C3 and subsequent deposition of the cleavage product C3b is necessary for the downstream conversion of C5. To further look into this, we examined phage activity in the presence of serum with the addition of specific complement inhibitors: compstatin, which inhibits C3 cleavage [41,42], and OmCI, which inhibits C5 cleavage [30] (Figure 3b). We observed that compstatin partially rescued phage activity in the presence of serum, while OmCI did not prevent inhibition by serum at all. Therefore, our results point to the involvement of the early stages of the complement system in the inhibitory effect of serum on phage PBJ. Notably, there was no difference in the activity of phages in the presence of heat-inactivated serum or heat-inactivated serum with complement inhibitors (Appendix A). However, as the addition of compstatin only partially recovered phage activity, we tested an additional panel of complement inhibitors, in this case nanobodies. Nanobodies are single-domain antibody fragments that bind antigens, with the advantage of having a much smaller size than an antibody [43].

We used nanobody C1qNb75, which inhibits activation of the complement classical pathway by binding C1q, the first step of the cascade [31]. In addition to this, we used nanobodies hC3Nb1 and hC3Nb2, which inhibit the cleavage of C3 in the alternative pathway or all activation pathways, respectively [32,33]. Finally, we used sdAb_E4, a C5 binder, to inhibit the terminal stage of the complement cascade [34]. We assessed phage activity in serum in the presence of the nanobodies at a range of concentrations. To compare their effects, we analyzed the influx of fluorescent DNA dye after 90 min of infection (Figure 3c,d). The C1q-targeting nanobody was able to recover phage activity in a concentration-dependent manner to the level of the control. Both of the C3-targeting nanobodies exhibited an intermediate effect, comparable to that of compstatin. As expected, the C5-targeting nanobody barely improved phage activity even at the highest concentration, which is in line with our observations with OmCI. Taken together, our results indicate that phage inhibition is likely caused by complement and more specifically by activation of the classical pathway.

### 3.4. Inhibition of Phages by Complement Occurs at the Stage of Phage Adsorption

One of the stages of phage infection that the complement system could be hampering is the binding of phages to their host. Phage adsorption can be measured with assays where bacteria are incubated with phages and the remaining free plaque-forming units are quantified. However, in such an assay, both phage adsorption and phage inactivation would translate to a decrease in free plaque-forming units. To circumvent this, we assessed phage adsorption by using fluorescently tagged phages.

Producing fluorescent phages poses a challenge. Random labeling strategies may compromise the activity of the phage as the dye has to react with the capsid and tail proteins [44]. We circumvented this by producing phages tagged with a chemical handle that can be functionalized through click chemistry [45]. To do so, we used a methionine-auxotrophic strain of *P. aeruginosa*, which requires exogenous methionine to grow. Methionine can also be partially substituted by its structural homologue L-azidohomoalanine, a non-canonical amino acid that contains an azide group. In a situation of methionine scarcity, bacteria will incorporate L-azidohomoalanine into their proteins. If phages are propagated with these bacteria as hosts, their proteins will also be azide-tagged (Figure 4a) [46]. The resulting virions are then available for biorthogonal labeling via click chemistry, as the azide groups can react efficiently with alkyne groups attached to a fluorophore to create a stable covalent bond. We amplified PBJ in this way and used the molecule DBCO functionalized with an Alexa-488 dye to fluorescently label our azido-tagged phages, obtaining PBJ-AF488.

We confirmed the labeling of PBJ-AF488 by means of widefield fluorescence microscopy. To do so, we incubated fluorescently labeled phages with PAO1 expressing cytoplasmic sfCherry (PAO1-sfCherry) to allow the phages to bind. We successfully detected bacteria with bound phages on their surface, as well as isolated phage clusters (Figure 4b). The addition of serum in this setting resulted in a decrease in the number of bacteria with attached phage particles (Figure 4c). Next, we used flow cytometry to confirm whether complement indeed hampers phage binding with a more quantitative approach. In this assay, we again used PAO1-sfCherry to facilitate gating (Appendix A). PBJ-AF488 or heat-inactivated PBJ-AF488 was incubated with bacteria for 5 min to allow for phage adsorption in the presence of serum or serum with inhibitors. Analysis of phage binding in these conditions revealed that the phage fluorescent signal increases when increasing the MOI in the absence of serum (Figure 4d). Although a relatively high MOI was necessary to obtain a sufficiently high phage-binding signal, it seems that phage binding was not yet saturated even at the highest MOI. Heat-inactivated phages did not bind to bacteria, supporting that the signal was specific for the binding of active phages. When serum was added, phage binding showed a sharp decrease (Figure 4d,e). As seen before in the phage lysis assays, the inhibitory effect became less pronounced as the MOI increased. In accordance with our previous results, we could recover phage binding almost fully by inhibiting serum with the C1q-targeting nanobody (C1qNb75), while the nanobody targeting all forms of C3 (hC3Nb2) only partially recovered phage binding. In all of the conditions analyzed, the shift in the phage fluorescent signal occurred homogeneously, meaning that phage binding was evenly distributed throughout the bacterial population (Figure 4e). In addition to phage binding, we stained the samples for bound C3b to confirm that there was complement activation on the bacterial surface. Very high levels of C3b were detected on bacteria incubated with active serum (Figure 4f,g). Interestingly, both nanobodies could completely inhibit C3b deposition at the concentrations tested. In the conditions treated with the C3 nanobody, phage biding was still inhibited even in the absence of C3b deposition. In conclusion, these results show that active complement, and in particular C1q, hinders the ability of phages to bind to their host.

### 3.5. Complement Inhibitory Effect Is Not Mediated by Receptor Blockage

Our results from phage binding assays performed with fluorescent phages reveal that complement blocks phage binding to bacteria. This could be due to complement components shielding receptors on the bacterial surface. However, it could also be caused by a direct interaction of complement with the viral particles. To discriminate which of these processes was hampering phage binding, we first studied the infectivity of phages on bacteria pre-opsonized with HPS. Opsonization was achieved by incubating bacteria with HPS for 30 min at 37 °C and washing away the unbound serum components. We confirmed that bacteria treated in this way were indeed covered with complement components C3b (Figure 5a). We then used the fluorescent DNA dye assay to assess whether phages induce the lysis of pre-opsonized bacteria (Figure 5b). When phages were added to the bacteria together with fresh buffer (RPMI), we still observed an increase in the fluorescence signal. In the case where phages were added together with fresh serum, phage infection was still inhibited, consistent with our previous findings. To address whether the inhibitory effect of complement is the result of an interaction with the phage, we performed an ELISA to assess the binding of purified C1q to phage particles (Figure 5c). This showed that C1q can indeed bind directly to immobilized phages in a concentration-dependent manner. This interaction could be responsible for blocking the ability of phages to attach to their host. In summary, our results indicate that phage inhibition is not mediated by complement blocking phage adsorption to receptors on the bacterial surface, but rather by an interaction of complement components with the viral particles.

## 4. Discussion

The past decade has seen an unprecedented increase in interest regarding bacteriophage therapy. New efforts are being made to investigate its clinical potential, while physicians strive to work around the stringent regulations attached to its use. At the same time, numerous case reports have been published in the past few years highlighting the potential of phages as therapeutics, both to substitute and complement antibiotics. In this context, it is important to keep advancing our fundamental knowledge on how phages may behave inside the body of a patient.

The purpose of our study was to explore the interactions of phages with the human immune system, and in particular with the complement system. To tackle this issue, we used human serum as a source of complement and characterized the performance of phages in this medium using different assays. Our results show that some phages are inhibited by serum in a concentration-dependent way, while others remain unaffected. When looking at a broader panel of phages, it was only the myophages that were challenged by this phenomenon. However, due to the limited number of podo- and siphophages available for us to test, it is difficult to fully relate the observation to the phage morphotype. Furthermore, we do not know the receptors of all the phages we tested, what their optimal host is, or how closely related they are to each other. Nonetheless, when we tested the activity of *S. aureus* myophage K, we saw that it was also inhibited by serum, further hinting at the importance of the morphotype. In any case, the examination of a more complete and well characterized set of phages, including more phages targeting other bacterial species, would be valuable in finding patterns to potentially predict sensitivity to serum.

We were also able to show that phage inhibition is mediated by the early stages of the classical complement pathway, as infective activity could be rescued by blocking C1q or C3. The action of the classical pathway, initiated by the deposition of C1q, can kickstart complement activation, while the alternative pathway can later on contribute to amplifying the reaction. This is reflected by our results with the nanobodies targeting C1q and C3. While both C3-targeting nanobodies could partially counteract the inhibitory effect of serum, it was only when C1q was also targeted that we observed a full recovery of the phage infection.

In this study, we also demonstrated a novel technique for producing fluorescently labeled phages in *P. aeruginosa*, based on incorporation of the non-natural amino acid L-azidohomolalanine and biorthogonal labeling by means of click chemistry. This method is more labor-intensive than random labeling approaches, but we found that it better preserved the activity of the phages. Furthermore, azido-chains incorporated into the phage capsid could potentially be functionalized with other molecules for a variety of applications. Using this technique, we were able to produce fluorescently labeled PBJ in high titers and study its binding to PAO1 through microscopy and flow cytometry. A high phage MOI was needed in these experiments to produce a detectable fluorescent signal. This constitutes a limitation, as the experimental conditions were not directly comparable to those of the other assays performed in this study. Nonetheless, the experiments described in Section 4 revealed that complement hampers phages by interfering with their ability to bind to the surface of the host. An intuitive explanation for this could be that C1q or C3b could shield phage receptors. However, when we assessed phage activity on bacteria opsonized with complement components using the fluorescent DNA dye assay, we observed that phages were still able to induce lysis. We hypothesize that complement activation may have a direct neutralizing effect on phages. The ability of complement to recognize and opsonize viral particles has been shown in studies with eukaryotic viruses [47], and the same could hold true for phages. Attachment of heavy proteins such as C1q or C3b to the viral particles could interfere with their infectivity. This could be particularly relevant for myophages, as they rely on the contraction of their tail to introduce their genetic material into the host [48]. Addressing these questions would be an interesting perspective for future studies.

Although we found that phage inhibition was not mediated by neutralizing antibodies in our assays, specific anti-phage antibodies in our serum pool could also be important in activating the classical complement pathway. Antibodies against head proteins of *Escherichia coli* (*E. coli*) myophage T4 have been found in individuals who have never been subjected to phage therapy [49]. The same proteins could induce the production of neutralizing antibodies and complement activation when used to immunize mice. Similarly, in a study performed on an *E. coli* podophage T7 peptide display library, recognition of certain peptides by IgM was seen to initiate a neutralizing complement response [50]. Nonetheless, our results show that C1q can bind directly to phage particles in the absence of antibodies. There is other evidence that activation of the classical pathway on viruses can occur independently of antibodies; for example, through the interaction of C1q with serum C reactive protein [51]. Additionally, direct binding of C1q to viruses such as dengue, influenza A, and SARS-CoV-2 seems to be sufficient in some cases to reduce infectivity [52,53,54]. Inactivation of phages by complement components independent of antibodies could explain the observations reported in several studies where phage activity was compromised in the presence of human serum or plasma [55,56,57].

Taken together, our results provide evidence that certain phages may be less competent than others in targeting bacteria in certain physiological situations, like in blood or tissues where complement is present. Although it remains unclear how these results would translate to an in vivo situation, our observations could still have implications on how we select phages for therapy. Researchers are working on ways to optimize this process and make it faster and more efficient. To this end, phage biobanks are currently being set up in different countries, and sharing phages between different institutions is gradually becoming a more usual practice. Several phages against *P. aeruginosa* have been characterized and are being used as medicinal products in the context of magistral phage preparations [58,59]. Efforts are also being made in engineering phages to increase their host range, as well as in optimizing phage cocktails to avoid the problem of phage resistance. Additionally, bioinformatic pipelines have been developed that may allow us to predict which phages will be more effective against a certain clinical isolate [60]. However, in spite of all these important advances, the interactions of phages with the patient’s innate immune system are still largely ignored in the process of therapeutic phage selection. In light of our findings, we suggest that performing screenings in the presence of serum would be valuable when designing phage cocktails.

In conclusion, this study shows that complement activation via the classical pathway can inhibit the anti-bacterial activity of certain phages, with myophages being more susceptible to this effect. We have shown that inhibition is not mediated by deposition on the bacterial surface but is rather caused by a direct effect on phages themselves. Our findings highlight the need to better understand how phages behave in physiological conditions when administered to a patient as a therapeutic. Selection of phages according to their expected performance in human blood or tissues could contribute to improving the efficacy of phage therapy.

## Figures and Tables

**Figure 1 viruses-15-02211-f001:**
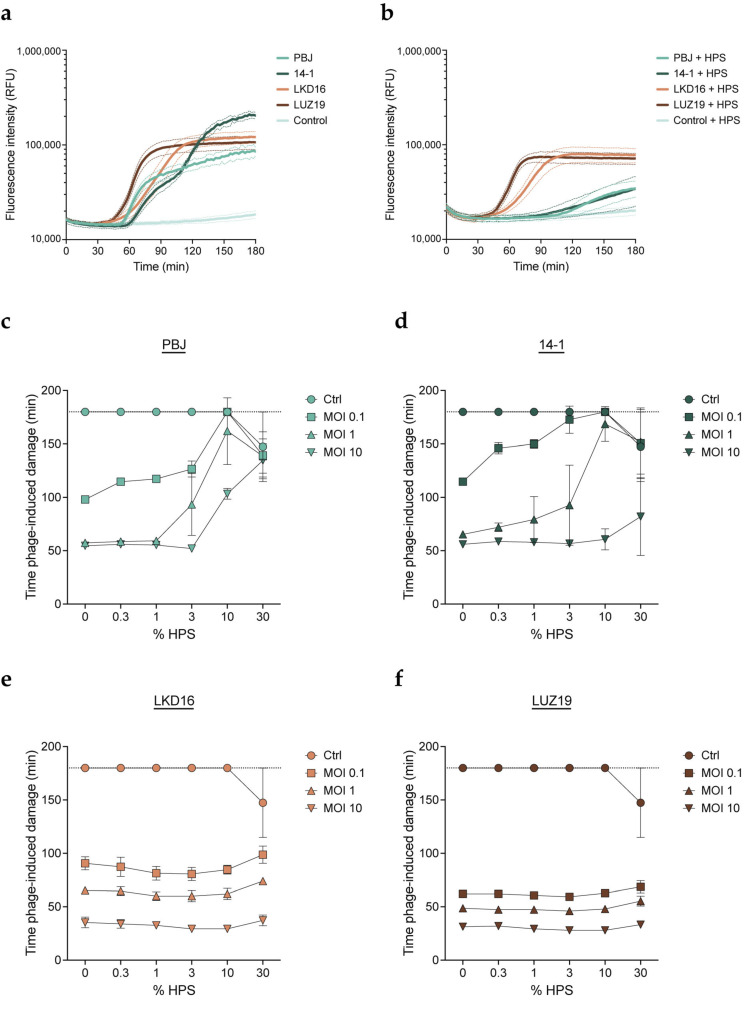
Human serum inhibits lytic activity of myophages. *Pseudomonas aeruginosa* strain PAO1 was incubated with bacteriophages (phages) PBJ, 14-1, LKD16, or LUZ19 at 37 °C in the presence of the DNA dye SYTOX Green. Phage-mediated damage is signaled by an increase in the fluorescence intensity of SYTOX Green. (**a**) Fluorescence intensity (relative fluorescence units, RFUs) over time of PAO1 infected with different phages at a multiplicity of infection (MOI) of 1 in the absence of serum. Uninfected bacteria were used as the control. (**b**) Fluorescence intensity (RFUs) over time of PAO1 infected with different phages at an MOI of 1 in the presence of 10% human pooled serum (HPS). The control is uninfected bacteria incubated with 10% HPS. (**c**–**f**) Time of damage induction (min) by phage (**c**) PBJ, (**d**) 14-1, (**e**) LDK16, and (**f**) LUZ19, at a range of MOIs (0.1, 1, 10) combined with HPS at different concentrations (0–30%). Time of damage induction is determined as the time after the addition of phages at which the fluorescence curve experiences a sharp and steady increase (1000 RFUs over the previous measurement). (**a**–**f**) Data represent mean ± SD of three independent experiments.

**Figure 2 viruses-15-02211-f002:**
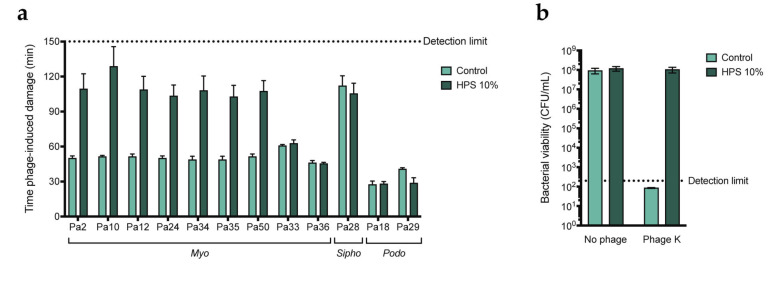
Human serum inhibits a variety of myophages targeting PAO1. (**a**) Time of damage induction (min) of the different phages as determined by the fluorescent DNA dye assay. Time of damage induction is determined here as the time after the addition of phages at which the fluorescence curve experiences a sharp and steady increase (700 RFUs over the previous measurement). Phages were added at an MOI of 1 and incubated with PAO1 at 37 °C in the absence of serum (control) or in the presence of 10% HPS. (**b**) *S. aureus* strain ATCC 19685 was incubated with phage K at an MOI of 10 in the absence (control) or presence of 10% HPS for 2 h at 37 °C, after which bacteria were plated and incubated overnight. Number of recovered colonies per plated volume is expressed as colony forming units per mL (CFUs/mL). Black dotted line represents detection limit. (**a**,**b**) Data represent mean ± SD of three independent experiments.

**Figure 3 viruses-15-02211-f003:**
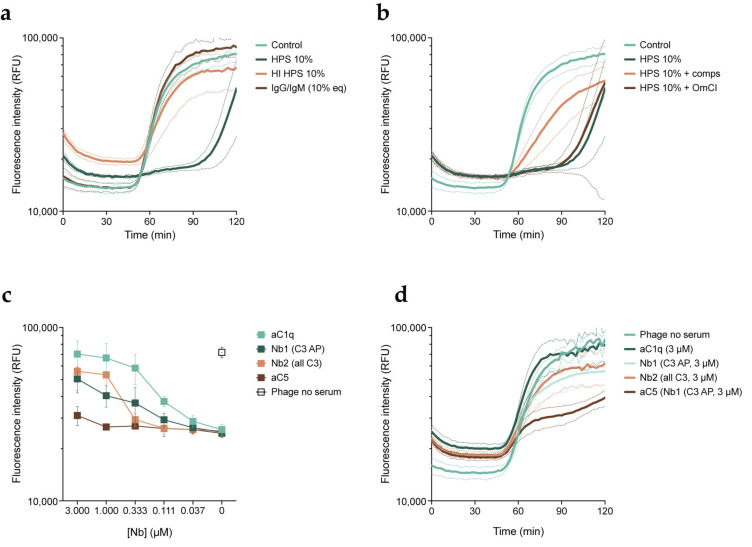
Inhibition of phages by serum is mediated by the complement system. PAO1 was incubated with phage PBJ at 37 °C in the presence of the DNA dye SYTOX Green. (**a**) Fluorescence intensity (RFUs) over time (min) of bacteria infected with PBJ at an MOI of 10 (control), in the presence of 10% HPS, 10% HPS with heat-inactivated complement (HI HPS), or IgG and IgM purified from HPS at a concentration equivalent to 10%. (**b**) Fluorescence intensity (RFUs) over time (min) of bacteria infected with PBJ at an MOI of 10 (control), in the presence of 10% HPS, 10% HPS with 50 μM compstatin, or 10% HPS with 20 μg/mL OmCI. (**c**) Fluorescence intensity (RFUs) of bacteria after 90 min of infection with PBJ at an MOI of 10 in the presence of 10% HPS and nanobodies at a range of concentrations. Value for bacteria infected with phage in the absence of serum is shown as a control. (**d**) Fluorescence intensity (RFUs) over time (min) of bacteria infected with PBJ at an MOI of 10 in the absence of serum (control) or in the presence of 10% HPS and nanobodies at a concentration of 3 μM. (**a**–**d**) Data represent mean ± SD of three independent experiments.

**Figure 4 viruses-15-02211-f004:**
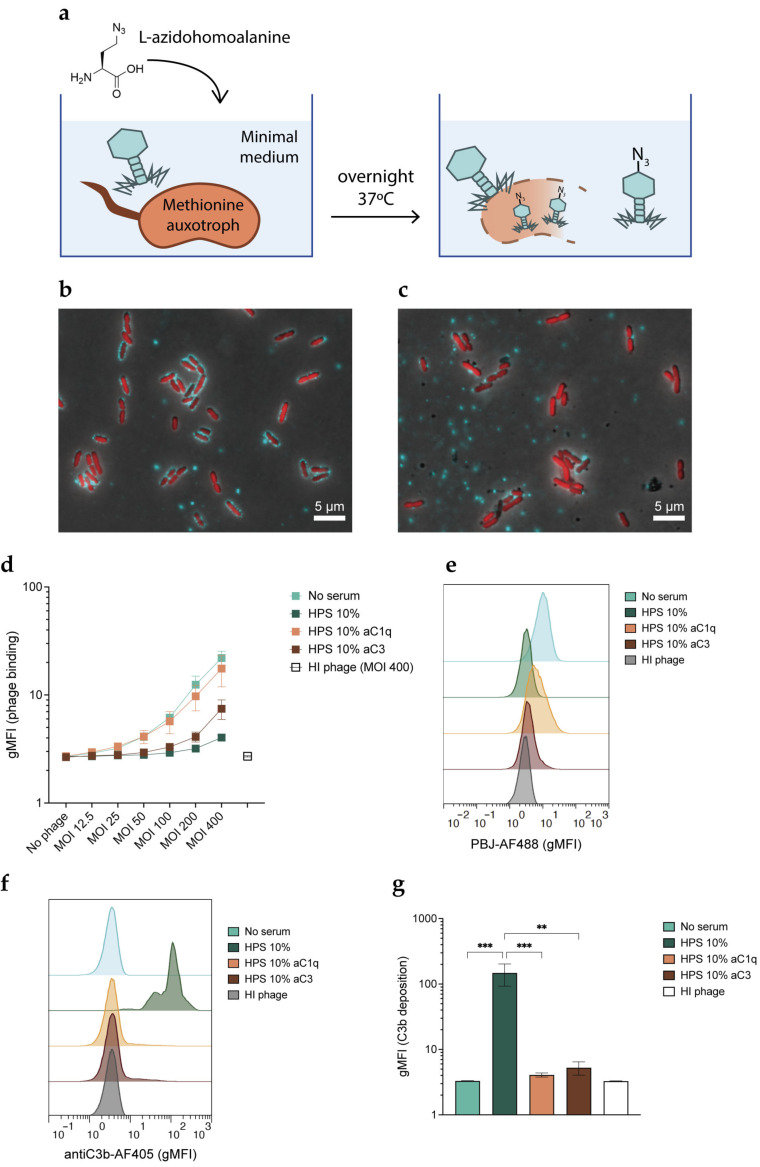
Phages tagged with an azide group and labeled with a fluorescent marker were used to study their binding to bacteria. (**a**) Schematic view of the technique used to produce azido-tagged phages. A methionine auxotroph PAO1 mutant was cultured in minimal medium supplemented with L-azidohomoalanine. Phages were amplified overnight at 37 °C using methionine auxotroph PAO1 as the host. The resulting phage progeny incorporates the non-canonical amino acid L-azidohomoalanine in place of methionine in its proteins, making them available for biorthogonal labeling via click chemistry. (**b**,**c**) PAO1-sfCherry (red) was incubated with PBJ-AF488 (green, MOI 50) for 5 min at 37 °C in (**b**) buffer or (**c**) 10% HPS, fixed in 1% PFA, washed, and immobilized on agar pads. Images were acquired with a 100× immersion objective and are overlays of the modes phase contrast and widefield fluorescence with cube filters. Images are representative of three independent experiments. (**d**–**g**) Flow cytometry was performed on PAO1-sfCherry incubated with PBJ-AF488 at different MOIs. When indicated, 10% HPS, anti-C1q nanobody (1 μM), or anti-C3 nanobody (1 μM) was added for 5 min at 37 °C. Samples were stained with monoclonal anti-C3-AF405 antibody and fixed in 1% PFA before measuring. Bacteria were gated on sfCherry-height and forward scatter-height signals. (**d**) Geometric mean of the signal corresponding to bound phages. Heat-inactivated (HI) phage was used as a negative control. (**e**) Histograms of phage fluorescent signal at an MOI of 200 or HI phage at an MOI of 400 in the various serum conditions. (**f**) Histograms of fluorescent signal corresponding to C3b deposition (AF405) on bacteria treated with the various serum conditions. (**g**) Geometric mean of the signal corresponding to C3b (AF405) for the conditions with PBJ-AF488 at an MOI of 200, or HI phage at an MOI of 400. Statistical analysis was performed using an ordinary one-way ANOVA with Tukey’s multiple comparisons test. Significance is shown as ** *p* ≤ 0.01 or *** *p* ≤ 0.001. (**d**,**g**) Data represent mean ± SD of three independent experiments. (**e**,**f**) Signal was normalized by number of events. A representative graph of three independent experiments is shown.

**Figure 5 viruses-15-02211-f005:**
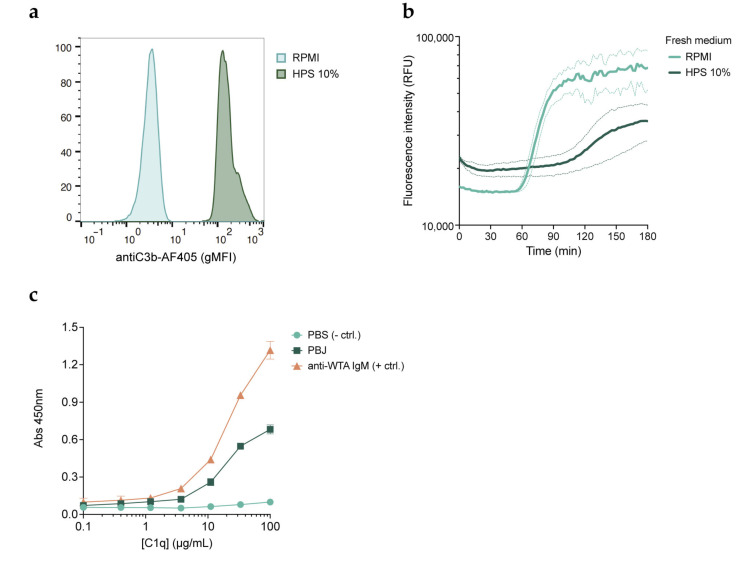
Inhibition of phages by the complement system is not caused by blockage of phage receptors. (**a**) Flow cytometry histograms showing C3b deposition (AF405) on bacteria incubated with 10% HPS for 30 min at 37 °C after removal of unbound serum components and staining with aC3b-AF405 nanobody. Signal was normalized by number of events. A representative graph of three independent experiments is shown. (**b**) Fluorescence intensity (RFUs) over time of complement pre-opsonized PAO1 incubated at 37 °C with phage PBJ (MOI 10) added either in buffer (RPMI) or in 10% HPS in the presence of SYTOX Green. Data represent mean ± SD of three independent experiments. (**c**) An ELISA was performed to assess the binding of C1q to phage PBJ. Wells coated with buffer were used as a negative control, while wells coated with an anti-*S. aureus* WTA human IgM were used as a positive control. Data represent mean ± SD of an experiment performed in triplicate.

## Data Availability

All the datasets generated and analyzed during this study are presented in the main text or the Appendix A. Genomic sequence of phage PBJ is available from Genbank, accession number: OR611941.

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
