# Peer review of "Human Complement Inhibits Myophages against Pseudomonas aeruginosa"

_viruses, 2023, doi:10.3390/v15112211_

Round 1

Reviewer 1 Report

Comments and Suggestions for Authors

The manuscript describes the influence of human serum complement on bacteriophage infection and demonstrate that human serum prevents infection of bacteria by myo- but not by podo- or siphophages. The authors show indirectly that C1q interacts with the myophages thereby plausibly blocking the phage adsorption. The authors demonstrate elegantly that blocking of C1q with anti-C1q nanobodies recovered the phage activity to infect the bacteria. This is an interesting finding and as a whole the manuscript is enjoyable reading.

Major comment:

1.      Experimentally the manuscript is only lacking the direct evidence that C1q binds to phages. The authors can only assume that C1q binds directly to phages as the phage receptors were not blocked on bacteria pre-incubated with human serum. The binding of C1q to phages should be experimentally verified. That should not be too difficult to test. A possible experiment to demonstrate this binding could be with ELISA: C1q coated wells, then incubated with fluorescently labelled phages, followed by MFI detection of the bound phages. The setup can also be turned around, so that the plate could be coated with purified bacteriophages, then incubated with C1q, followed by detection with anti-C1q-antibody.

Minor comments:

2.      L166, section 2.6. … phage concentration of … BACTERIOPHAGES/mL

3.      L174. … anti-C3) were ADDED when indicated

4.      L195-196. Should this be agarose instead of agar?

5.      Section 2.9. Which tool was used to draw the figures?

6.      Result 3.5 could be joined with result 3.2 for better clarity. Result 3.2 alone is slightly confusing.

7.      Figure 4g: A statistical test between the groups would be good.

Reviewer 2 Report

Comments and Suggestions for Authors

In the manuscript “Human complement inhibits myophages against Pseudomonas aeruginosa” authors study the effects of serum components (antibodies and complement proteins) on the activity of phages of different morphology.

This study is well-planned, methods are proper (in general) and sufficient; the discussion is clear. The manuscript is well-written. I believe that the manuscript is of interest for the readers of the journal.

I have no major comments, but one comment and several notes.

The comment:

In 3.3., the authors include the results of experiments on the role of serum antibodies (non-purified from serum and purified IgG and IgM) on the possible inactivation of phages. This part of 3.3 does not make sense. First, the authors have not proved that there are specific antibodies in the used pooled serum (western blot, please). And second, if specific antibodies are present in HPS, being non-neutralizing, they should not affect the inactivation of phages. So, it is better to remove this part from 3.3., since the rest of the chapter adequately and accurately describes the experiments that are necessary for the conclusion.

 Notes:

1.       Why do the authors write “The genomes were assembled …” (L 128) if they prepared and sequenced one phage genome DNA library (p 2.2). Please, correct.

2.       The same phrase. What does it mean “… including –careful[28] (L 129). Please, complete.

3.       Although the authors indicate that 14-1, LKD16, and LUZ19 phages have been extensively characterized and provide references (L 213-214), it would be useful (and friendly for the readers) to add their taxonomic position.

 I would recommend the author to make the corrections.

Comments on the Quality of English Language

The English is good enough

Round 2

Reviewer 1 Report

Comments and Suggestions for Authors

The authors have revised the manuscript as requested